# Personal response to immune checkpoint inhibitors of patients with advanced melanoma explained by a computational model of cellular immunity, tumor growth, and drug

**D. Perlstein[1]⊙, O. Shlagman[1]⊙, Y. Kogan[1], K. Halevi-Tobias[1], A. Yakobson[2], I. Lazarev[2], Z. Agur(iD)[1]***

**1** Institute for Medical Biomathematics (IMBM), Bene Ataroth, Israel, **2** Department of Oncology, Soroka University Medical Center, Be'er Sheba, Israel

⊙ These authors contributed equally to this work.
* agur@imbm.org

## Abstract

Immune checkpoint inhibitors, such as pembrolizumab, are transforming clinical oncology. Yet, insufficient overall response rate, and accelerated tumor growth rate in some patients, highlight the need for identifying potential responders. To construct a computational model, identifying response predictors, and enabling immunotherapy personalization. The combined dynamics of cellular immunity, pembrolizumab, and the melanoma cancer were modeled by a set of ordinary differential equations. The model relies on a scheme of T memory stem cells, progressively differentiating into effector CD8+ T cells, and additionally includes T cell exhaustion, reinvigoration and senescence. Clinical data of a pembrolizumab-treated patient with advanced melanoma (Patient O') were used for model calibration and simulations. Virtual patient populations, varying in one parameter or more, were generated for retrieving clinical studies. Simulations captured the major features of Patient O's disease, displaying a good fit to her clinical data. A temporary increase in tumor burden, as implied by the clinical data, was obtained only when assuming aberrant self-renewal rates. Variation in effector T cell cytotoxicity was sufficient for simulating dynamics that vary from rapid progression to complete cure, while variation in tumor immunogenicity has a delayed and limited effect on response. Simulations of a-specific clinical trial were in good agreement with the clinical results, demonstrating positive correlations between response to pembrolizumab and the ratio of reinvigoration to baseline tumor load. These results were obtained by assuming inter-patient variation in the toxicity of effector CD8+ T cells, and in their intrinsic division rate, as well as by assuming that the intrinsic division rate of cancer cells is correlated with the baseline tumor burden. In conclusion, hyperprogression can result from lower patient-specific effector cytotoxicity, a temporary increase in tumor load is unlikely to result from real tumor growth, and the ratio of reinvigoration to tumor load can predict personal response to pembrolizumab. Upon further validation, the model can serve for immunotherapy personalization.

**Data Availability Statement:** All relevant data are within the paper and its Supporting Information files.

**Funding:** The author(s) received no specific funding for this work.

**Competing interests:** The authors have declared that no competing interests exist.

# Introduction

Currently, melanoma is the fifth most common cancer in men and the sixth most common in women in the US [1], 5-year survival rate being 15–20% for stage IV disease [2]. This state hopefully changes now, since over recent years melanoma treatment has been revolutionized with the approval of tyrosine kinase inhibitors and immune checkpoint inhibitors (ICI), both showing a significant impact on prognosis [3–5]. In particular, pembrolizumab, the inhibitor of the immune checkpoint receptor Programmed cell Death 1 (PD-1), manifested unprecedented efficacy in advanced melanoma patients, showing 33% objective response rate, 8.9% complete response (CR) rate, and progression-free survival of 6.9 months [6]. Accordingly, in 2014, pembrolizumab was approved for clinical use in the US and other countries [5, 7].

However, even though the application of PD-1 inhibitors is a remarkable breakthrough in modern oncology, it involves complications, such as immune-related adverse events, or new and unexpected types of response [8, 9]. Some melanoma patients (circa. 12%; [10]) show greatly accelerated tumor growth accompanied by clinical deterioration—a phenomenon known as Hyperprogressive Disease (HPD). HPD is an increasingly recognized response, for which positive predictors, other than, possibly, advanced age, are not yet identified. This raises a serious concern about the treatment of certain patient groups, calling for urgent investigation of the underlying mechanisms [11, 12]. Another atypical response, termed pseudoprogression, is an early or delayed increase in tumor load (>25%), which is not confirmed as a progressive disease in the following assessments. This phenomenon may be due to the infiltration of cytotoxic T cells into the tumor, edema or necrosis, or due to true short-term tumor growth in tumor load [13]. About 7% of pembrolizumab-treated advanced melanoma patients manifest early or delayed pseudoprogression, resulting in recommendations to apply ICI beyond the radiographically confirmed progression [5]. As ICI become widely available, the ability to foresee HPD or to differentiate pseudoprogression from real progression may be critical in avoiding either rapid disease acceleration or premature withdrawal of treatment. How to predict atypical responses of a patient and adjust treatment plans accordingly is a big challenge in the current immunotherapy practice [14].

The effect of ICI is governed by complex interactive dynamics of the patient's immune system, the growing tumor, and the ICI drug. In pathogenic diseases in humans, T lymphocytes expand extensively upon encountering foreign antigens, and following the peak of expansion, the resolution of the inflammation and the clearance of the antigen, this expanded T cell population undergoes apoptosis. The mechanism underlying this immune regulation involves two main players, PD-1—an immunoinhibitory receptor, expressed on T and B cells, and its ligand —programmed cell death ligand (PD-L) 1, expressed on T cells, B cells, macrophages, dendritic cells, and nonimmune cells. The expression of PD-L1 and the subsequent binding to PD-1 induces T cells to undergo apoptosis, consequently containing the expansion of the immune response. Melanoma cells can hijack this mechanism by expressing the same co-inhibitory signal, PD-L1, within the tumor microenvironment. These PD-L1 molecules bind to PD-1 receptors on T lymphocyte, stimulating these cells to undergo apoptosis, weaken the immune response prematurely, and hamper cancer cell clearance [15]. The ICI drugs were engineered to disrupt this cancer-induced ligand–receptor association and premature apoptosis. The humanized antibody drug, Pembrolizumab, for example, blocks the PD-L1, 2 inhibitory pathway. As a result, exhausted T cells can be re-activated, become more efficient in tumor surveillance, and enable the restoration of anticancer immunity and the suppression of cancer growth [16].

Mathematical modeling is a powerful tool for succinctly describing complex biological systems and for examining the relative influence of various biological factors on the overall

dynamics. Mathematical models in immunology and immune-oncology have been developed for exploring dynamic properties of CD8$^+$ T cells, their memory, their responses to pathogens, how they can be boosted, and the conditions under which they contribute to the protection, (e.g., [17]). These models also can be instrumental in the design of immunotherapy protocols or the personalization of cancer immunotherapy [18–20]. However, the research in immune-oncology rapidly evolves, changing the ways the immune system is looked upon and prescribing new modeling concepts. The recently emerging perception of cellular immunity as a stemness-to-exhaustion continuum [21–26] challenges the suitability of previous models of cellular immunity. The landscape of cancer drug therapy is also changing, at present, highlighting the superior efficacy of the newly discovered PD-1/PD-L1 ICI. The ICI-driven dynamics must be incorporated in the mathematical models describing responses of cancer patients to this immunotherapy modality.

In this work, we have developed a mathematical model describing the combined interactions of the major components of the system. This model enables the analysis of the yet unexplained phenomena associated with responses of patients to ICI [18, 20], which would otherwise be unattainable due to the inherent system complexity. Our model describes the interactions among three forces: the cellular immune system—taken as a progressively differentiating tissue [23–25], the melanoma cancer, and the immunotherapy by PD-1/PD-L1 blockade. By incorporating two important regulatory mechanisms—cell senescence and lymphocyte exhaustion, the model is expected to account for a plethora of response patterns in patients receiving ICI treatment, and to pinpoint system parameters, which may aid in the search for new response markers to immunotherapy drugs.

The following rationale guided our work. First, we constructed a mathematical mechanistic model accounting for the above-described interactive dynamics. This model was then adjusted to fit the clinically measured dynamics of a reference patient from a hospital cohort, suffering from metastatic melanoma (Patient O'). This process involved evaluation of the patient's model parameters, based both on literature and the patient's clinical information. To examine the sensitivity of the patient response to potential inter-patient variation in specific parameters, we simulated the model under variation in each one of several model parameters.

The virtual, or *synthetic*, patient population approach, of creating a virtual patient model and then simulating clinical trials in populations of such virtual patients, was introduced by Agur and colleagues in early 2000s [27–30], and by Bangs [31]. According to this approach, a virtual patient is a collection of mathematical models formalizing the pertinent pathology or physiology. A virtual patient-population is a collection of virtual patients, each characterized by a set of model parameters drawn from the distributions of these parameters in the real patient-population. Once a virtual patient population is established it can undergo virtual clinical trials, endpoints of which being those employed in the pharmaceutical industry, such as Progression-Free Survival (PFS), Objective Response Rate (ORR), etc. [32]. The concept of virtual clinical trials was employed, e.g., for examining how prematurely shelved drugs can be rescued. Studying a discontinued drug, ISIS-5132, the virtual clinical trial analysis showed that by combining this molecule with a licensed drug, sunitinib malate (Sutent®, Pfizer Inc), the treatment of prostate cancer could be improved, with more patients reaching PFS at five years, as compared to either ISIS-5132 or sunitinib malate monotherapy [28]. In the present work, we used the understanding gained thus far to construct virtual patient populations, having inter-patient variation in one model parameter or more. This was done for retrieving results of a clinical trial, suggesting that the response to pembrolizumab is associated with the reinvigoration-to-tumor load ratio [33]. The ability to retrieve a substantial clinical result serves as a preliminary proof of the suitability of our model.

## Materials and methods

### Patient

We report a case of a 27 years old female with histology of primary thymic malignant melanoma, who was treated at the oncology department at Soroka hospital, Israel. The patient received treatment according to the therapy options for patients with melanoma, which were approved by the Israel Ministry of Health, and are included within the treatments reimbursed by the National Health Insurance. The patient underwent excision of the anterior mediastinal tumor following adjuvant radiation therapy with a total dose of 50 Gy in 25 fractions. Immunotherapy with ipilimumab was administered a month later when the metastatic disease was detected. In December 2014, following the second cycle of treatment, imaging showed progression and treatment was switched to pembrolizumab, 120 mg, every 3 weeks, by intravenous infusion over 30 minutes. Under pembrolizumab the patient responded and is still presently treated, manifesting low to negligible tumor load [34].

### Mathematical modeling

Our model incorporates the dynamic interactions of the cellular immune system, the progressing tumor, and the ICI, pembrolizumab (see graphical representation in Fig 1A).

**Tumor growth.**   We assumed that in the absence of inhibition by immune cells, the tumor grows according to a power-law function at a rate higher than linear, yet lower than exponential (exponent equaling 2/3). This assumption was validated by *in vitro* and *in vivo* results in several cancer indications. In particular, large scale mammography screening trial data in breast cancer demonstrate power-law growth (exponent approximating 1/2), but show inconsistency with both Gompertz and logistic growth functions [35]. The assumption was prospectively validated by one-, two- and three-dimensional tumor growth experiments both in vitro, in MCF-7 cells (breast cancer cell line) and in vivo, in mouse xenografts. In all studied cases, the unsaturated growth follows a power-law function with exponent 2/3 in the three-dimensional case [36]. In another work, the power-law growth assumption was tested experimentally in implanted human ovarian carcinoma spheroids. Results show that initially, similar tumor spheroids varied widely in their growth-rates, ranging from almost exponential growth to power-law growth with a smaller exponent [37]. In a work on hormone-sensitive prostate cancer (HSPC), a model assuming tumor growth law with a dynamic power successfully retrieved the individual disease time-courses of 83 HSPC patients treated by androgen deprivation therapy. Simulations of other tumor growth functions showed a less accurate fit to the clinical patient data [38]. Taken together, these experimental and theoretical results, in different solid tumor indications and states, suggest that the power-law is a general function describing intrinsic solid tumor growth.

**Dynamics of the cellular immune arm.**   In our model (see illustration in **Fig 1A**), we assume that antigen-specific naïve T cells (N) differentiate into stem cell memory (SCM) cells upon activation, giving rise to central memory (CM) cells. The latter differentiate into effector memory (EM) cells which further differentiate into effector (E) cells [23], and eventually into fully exhausted (EXH) cells. Tumor immunogenicity is represented by the basic rate at which dendritic cells (DCs) are stimulated by cancer cells to mature and home to sentinel lymph nodes, where these cells present cancer antigen to CD8$^+$ T cells, initiating the activation of tumor-specific CD8$^+$ T cells. We assume that N cells in the blood are sufficiently abundant to allow unrestricted differentiation into SCM cells. We further assume that cancer-activated DCs stimulate the SCM and CM populations to increase their cell division rates [22, 24, 26, 39] (here and below, the terms 'division', 'cell-division' and their derivatives refer to the discrete division

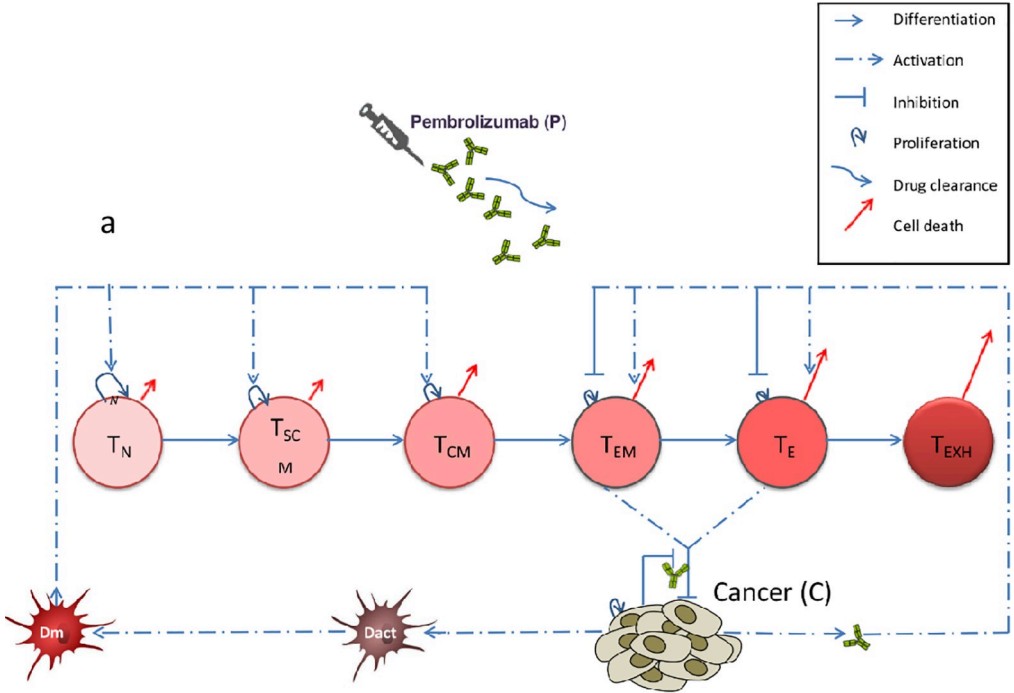

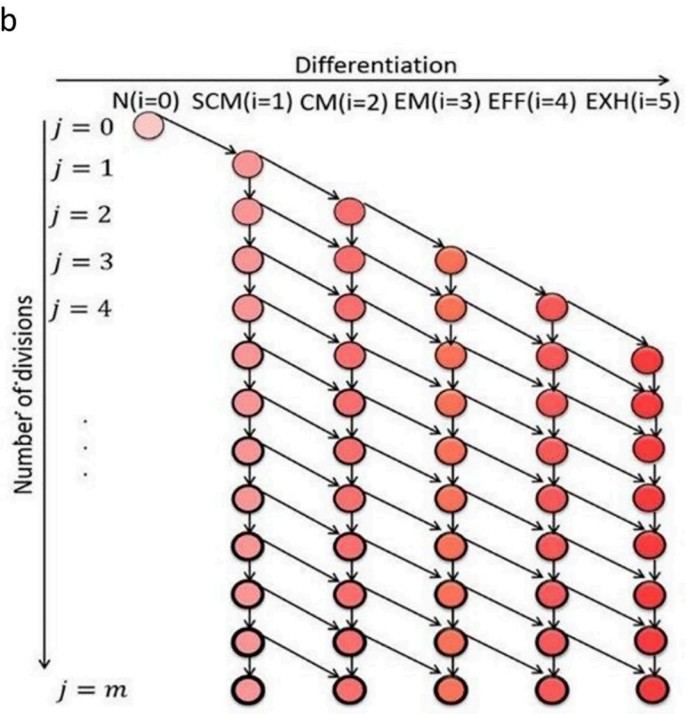

**Fig 1. A graphical representation of the mathematical model.** (1a) The interactions between the cellular immune arm, growing cancer and the immune checkpoint inhibitor, pembrolizumab. In our model dendritic cells (DC) are activated by cancer antigens ($D_{act}$, brown stain-like shape). They migrate to a nearby lymph node and become mature dendritic cells ($D_m$, red stain-like shape). Upon activation, naïve T cells ($T_N$) differentiate into T memory stem cells ($T_{SCM}$), which differentiate into central memory T cells ($T_{CM}$). The latter differentiate into effector memory T cells ($T_{EM}$), which in turn differentiate into Effector cells ($T_E$), and these gradually differentiate into fully exhausted T cells ($T_{EXH}$). This is depicted, from left to right, by increasingly dark red circles, color denoting increasing functionality. $D_m$ cells stimulate cell division of both $T_{SCM}$ and $T_{CM}$ cells, (blue dashed-dot arrows). Cells in each compartment self-renew (curled arrows; relative size representing the relative rate). Cells in each compartment differentiate (horizontal arrows) and die (red arrows; relative size representing the relative rate); for simplicity, the division is assumed to be symmetric. The PD-L1$^+$ cancer cells, C, affect $T_E$ and $T_{EM}$ cells via binding of PD-L1 to PD-1 receptors, generating inhibiting and activating signals: inhibition of cell division capacity and effector functionality (blue T-shaped lines), and activation of apoptosis (blue dashed-dot arrows). The killing rates of cancer cells by effector memory and effector cells are also reduced due to PD-1/PD-L1 ligation, as indicated by the T-shaped arrows. After being injected, the drug pembrolizumab (small, green, y-shaped objects) blocks the PD-1/PD-L1 pathway and prevents inhibition of $T_{EM}$ and $T_E$. (see formulation and further details in Supporting Information). (1b) Differentiation and senescence of T lymphocytes. Following each cell division, the daughter cell may remain in the same developmental compartment, (i, j+1; described by a vertical arrow), or advance to the next developmental compartment (i+1, j+1; described by a diagonal arrow). In the circles, the shades of red and the circumference lines represent the differentiation stage (going from naïve T cells in the lightest shade, to the most differentiated fully exhausted cells in the darkest shade, increasing senescence being shown by increasing thickness of the circles), and the expression of inhibitory receptors (increasing with the cell division index), respectively.

of a mother cell into two similar daughter cells, which can be either identical to the mother cell or differentiated). Successive T-cell developmental compartments are characterized by increasing cytotoxicity and by decreasing the replication rate [23]. Based on experimental findings, we assume that the PD-1 receptors are expressed in significant quantities on EM and E cells only, with greater abundance on the latter [9, 40, 41], and that PD-1 receptors bind to PD-L1 ligand on tumor cells to generate two inhibitory signals, which lead to exhaustion of CD8$^+$ T cells [42–44]. One of these signals impairs effector functions [45] while the other leads to apoptosis.

**Senescence.** We also include in the model the effects of senescence on the cellular immune arm [46], e.g., the aging of immune cells at each cell replication due to telomere shortening (Fig 1B). We adopt an approach similar to that introduced in [47] for modeling replicative senescence in the CD8$^+$ T-cell development. As illustrated in Fig 1B, each immune subset is indexed by a differentiation index, *i*, and a senescence index, *j*. The maximum number of divisions in a cell lineage, *m*, is arbitrarily set to *m = 25* and the cell division of T cells is assumed to decrease with increasing differentiation index [48].

**Immunotherapy by the ICI, pembrolizumab.** Our immunotherapy model assumes that some T cells have their PD-1 receptors blocked by anti-PD-1 antibodies, i.e., by the drug pembrolizumab, and therefore do not receive any apoptotic signal from PD-L1 on tumor cells. Thus, the cancer-mediated decrease in their effector functions (longevity, division capacity, and cytotoxicity) is partially annulled and their effector functions are restored. The mathematical model is fully presented in the Supporting Information section.

**Simulations of the mathematical model.** The mathematical model was numerically solved by the ode15s function of Matlab R2016a. To fit the model to the patient's data, we assumed initial conditions of uniformly distributed cells in all differentiation and senescence compartments. To guarantee that disease progression always takes place for untreated patients, all simulations of individuals treated by pembrolizumab were preceded by simulations of the same patients under no treatment.

## Fit of the model to Patient O's clinical data and parameter estimation

To evaluate realistic ranges of model parameters, we took as a reference a single patient, Patient O', who showed complex metastatic cancer behavior under immunotherapy,

constituting a good case study both medically and in the context of mathematical modeling. Clinical information about tumor load dynamics of this patient enabled us to evaluate her model parameters. In the first stage, we crudely evaluated most of the model parameters, based on information in the literature (to be denoted general parameters; see a summary of quantified parameter values, quantification methods and references in Table B in S1 Text. Other model parameters were not reliably quantifiable based on the literature, due to lack of relevant experimental or clinical information, or since they represent complex underlying processes, which cannot be evaluated from real-life information. The latter parameters were denoted specific parameters, as we assumed they are more patient-specific–an assumption to be studied below here. These parameters include the instantaneous rate of increase of cancer cells—$p_C$, baseline cytotoxicity of effector CD8+ T cells—$k_E^0$, tumor immunogenicity—$\rho_D$, the reinforcing effects of the tumor on effector memory and effector T cell inhibitions—$k_C^{03}$ and $k_C^{04}$, respectively, and the parameters defining the probability of self-renewal in the different T cell compartments − $a_{SCM}$, $a_{CM}$, $a_{EM}$ and $a_{EFF}$. To fine-tune the general parameters and to evaluate the specific parameters of Patient O', a fit was performed to best match the simulation results to the observed tumor load (defined as the sum of tumor lesions volumes) of Patient O', which was clinically monitored during over forty months of therapy. Each general parameter was allowed to vary within a narrow range, whose average is the value obtained based on the literature, while specific parameters were allowed to receive values selected from a relatively wide, conceivable, range. A cost function, quantifying the accuracy of fit, was defined based on the calculation of the overall root mean square error between data points and model predictions. The cost function additionally included penalties for infeasible trends and parameter values. Accordingly, a best fit was obtained using a custom-made optimization algorithm, which we tailored to efficiently search for minima in high-dimensional parameter space. The algorithm combines various schemes and search methods, including, among others, gradient descent, Markov Chain Monte Carlo employing importance sampling, genetic algorithm, and simulated annealing. The algorithm maintains a balance between global exploration of the entire parameter space, and focus on locally found minima. It proved marginally more efficient than Matlab's relevant built-in optimization functions (fmincon, ga, sa). Following the fitting procedure, a sensitivity analysis was performed, accompanied by a calculation of the associated confidence intervals, based on [https://stat.ethz.ch/~stahel/courses/cheming/nlreg10E.pdf].

## Generation of virtual patient populations

We tested the feasibility of our model by attempting to retrieve clinically-obtained results by Huang et al. [33], which suggest a positive correlation between a patient's response to ICI administration, and his or her ratio of reinvigorating T cells to baseline tumor load. To simulate the experiment in [33], we generated a large virtual patient population and selected 125 of them as a reference population, using a scheme to guarantee that each selected member has a tumor that progresses with no treatment (for further explanations, see below and the Supporting Information file). Members in the virtual population were simulated for 12 weeks during which pembrolizumab was administered at the beginning of the simulated period, and every three weeks thereafter. The disease state of each virtual member, after 12 weeks of treatment, was defined according to the Immune-related Response Evaluation Criteria in Solid Tumors (irRECIST), as done by Huang et al.[33]. The response was accordingly divided into complete response (CR; not achieved in the current virtual population results), partial response (PR), stable disease (SD), and progressive disease. Consistent with [33], members exhibiting the first two response types, were grouped as 'responders' and the others as 'non-responders'. Four virtual populations were constructed, and in each one of them the values of one parameter, or

more, were chosen to vary between patients. For each patient in each population, these values were randomly selected from a range of 0.28 to 3.5 fold of the heuristically determined base value for Patient O' (see above), except for the CD8+T cells division rates, which received values ranging from 0.5 to 2 fold of the literature derived values (see Supporting Information).

### Calculation of the reinvigoration level

As Ki67 is a marker for all dividing T cells, we evaluated in our simulations the percentage of cells bearing this marker by summing the numbers of dividing cells at any given moment in all lymphocyte compartments. The number of T cells undergoing reinvigoration was accordingly calculated as follows: i) multiply the number of T cells in each sub-compartment by the current cell division rate in that sub-compartment; ii) sum the products; iii) divide the sum by the total current number of T cells; iv) multiply the result by a time constant $\tau$, taken to be 24 hours, representing the average overall division time of a cell.

## Results

### Fit of the model to clinical data of patient O'

Fig 2 shows the clinically measured tumor load in one patient from a hospital cohort having advanced melanoma (Patient O'), during treatment by pembrolizumab. Also shown in this figure are the best-fit tumor load dynamics obtained by model simulations. One can see in **Fig 2** that the simulated tumor dynamics successfully captured the complex disease profile of patient O', including two alterations in the trend of her tumor load dynamics over three years of immunotherapy. Based on this fit, we estimated the values of all parameters of Patient O's model and analyzed the robustness of the evaluated parameters. Our analysis, combined with the exhaustive coverage of the biologically feasible parameter ranges, points towards the existence of a single global minimum for the specific parameter values of patient O'. In other words, the specific parameters evaluated for the patient are the most plausible ones within a conceivable range (See Materials and Methods, the caption 'self-renewal', as used herein, refers to the division of a mother cell into two daughter cells, wherein the daughter cells do not differentiate in the process, and are thus substantially identical to the mother cell [49].

A 90% confidence interval for the specific parameter, $p_c$, is obtained in the interval (0.978, 1.062) of its optimized value, obtained by the search. The sensitivity analysis showed a distinct optimum for the specific parameter, $k_C^{03}$, since the system reaches saturation above the optimized value. Accordingly, the likelihood function receives a constant value for all values above the optimized value, and therefore the appropriate definition for a confidence interval, in this case, would be the range between the optimized value and the upper parameter range limit, which was determined as described in the Supporting Information.

### Effect of parameter variation on response to treatment by pembrolizumab

The parameter values of Patient O's model served as the baseline values for investigating the effects of inter-patient variation in model parameters on the response to pembrolizumab. From all model parameters, we singled out those assumed decisive in determining the patient response, namely, the cancer cell replication rate—$p_C$, the baseline elimination of cancer cells due to effector CD8+ T cells—$k_E^0$, the tumor immunogenicity—$\rho_D$, the reinforcing effects of the tumor on effector memory and on effector T cells—$k_C^{03}$, $k_C^{04}$, respectively. To check our assumption that these parameters can determine the response pattern, we varied the values of each of them within a certain range and simulated the disease dynamics in the patients. As a control, we simulated untreated patients whose parameter values varied in the same respective

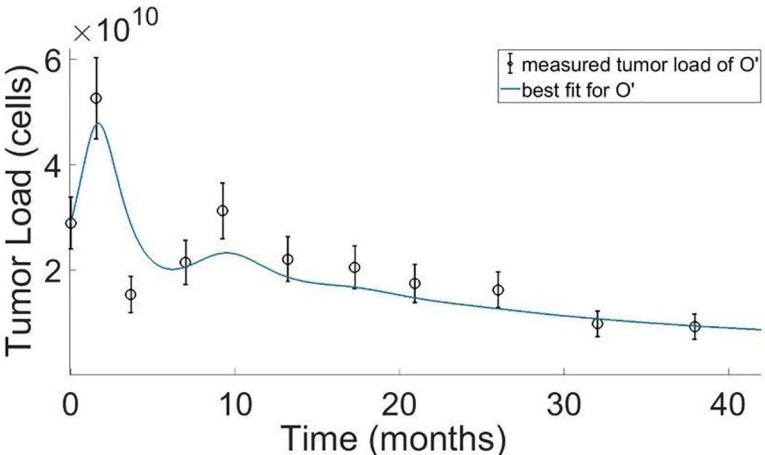

**Fig 2. Model fit to clinical data.** Clinical tumor load measured by PET/CT, derived from the clinical lesions data of an advanced melanoma patient from a hospital cohort (Patient O'), treated by pembrolizumab, 120 mg, every 3 weeks (empty circles). The data error bars were obtained assuming identical errors in measurements along all axes and given a maximal error of 3 millimeters for a standard CT measurement in each axis. The simulated time course of tumor load shows the best-fit to the clinically measured tumor load (curve). Model parameters of Patient O', evaluated from the best-fit simulation curve are: $p_C = 1.828$ h$^{-1}$, $k_E^0 = 0.00603$ h$^{-1}$, $\rho = 0.95$ h$^{-1}$, $k_C^{03} = 0.0058$ h$^{-1}$, $k_C^{04} = 0.00535$ h$^{-1}$, $a_{SCM} = 0.05$, $a_{CM} = 0.9$, $a_{EM} = 0.03$, and $a_{EFF} = 0.92$. For other model parameters see the Materials and Methods section and the Supporting Information file.

ranges; results consistently showed significant disease progression in all the untreated virtual patients (not shown).

The parameter $k_E^0$ represents the cytotoxicity of effector CD8+ T cells to cancer cells, before any loss of function due to exhaustion. We simulated tumor growth for different $k_E^0$ values, for patients treated by pembrolizumab, 120 mg, every 3 weeks for 28 months; all other parameters were kept constant. To examine the change in cancer dynamics as a function of the change in the value of $k_E^0$, the value for $k_E^0$ obtained for patient O' was iteratively multiplied by 1.05 (representing a 5% increase between successive sampled values) ten times, and in addition, it was subsequently divided by 1.05 ten times, to thereby obtaining a $1.05^{10} = 1.63$-fold range above and below the estimated baseline value for Patient O'. Varying $k_E^0$ in a 1.63-fold range above and below the estimated baseline value for Patient O', we noted that small differences in the baseline cytotoxicity of effector cells resulted in large variations of tumor dynamics during pembrolizumab treatment (Fig 3A). For small values of $k_E^0$ ($k_E^0 < 0.0045$ cells/hour), the model predicts a monotonic, steep increase in tumor load, to about 6-fold above baseline. The gradual rise of $k_E^0$ above 0.0045 cells/hour, moderates the increase in tumor load. A further rise in the values of $k_E^0$ changes the patient's response to SD, and eventually to CR.

Tumor immunogenicity, $\rho_D$, denotes the intrinsic rate at which DCs are stimulated by cancer cells. We simulated tumor growth for different values of $\rho_D$, following treatment by pembrolizumab, 120 mg, every 3 weeks, for 28 months; all other parameters were kept constant, as determined above for patient O'. Fig 3B shows model simulations of tumor load time-course for patients varying only in $\rho_D$. In Fig 3B one can note that in the first year of treatment, variations in $\rho_D$ affected disease progression to a relatively small degree, as compared to the effects of other specific parameters. During the subsequent years, increasingly growing differences in tumor load are expected, caused by the different levels of tumor immunogenicity, $\rho_D$. The differences in disease state for different levels of tumor immunogenicity became more dramatic after about three years of treatment, now ranging from cure to galloping progressive disease.

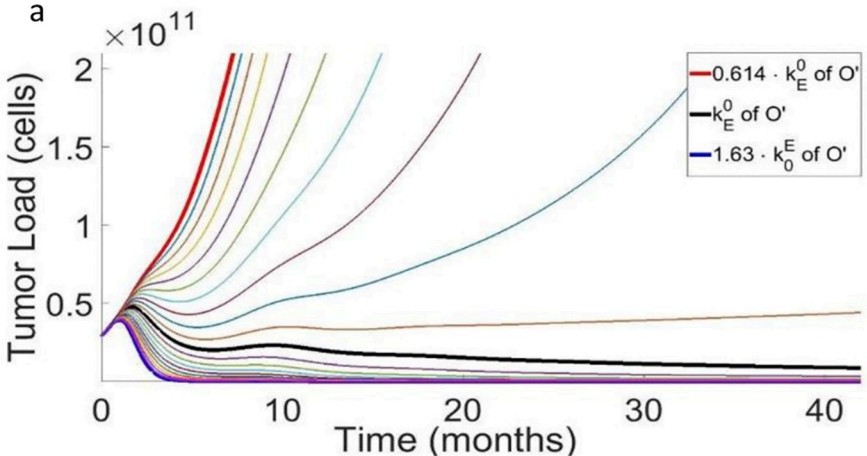

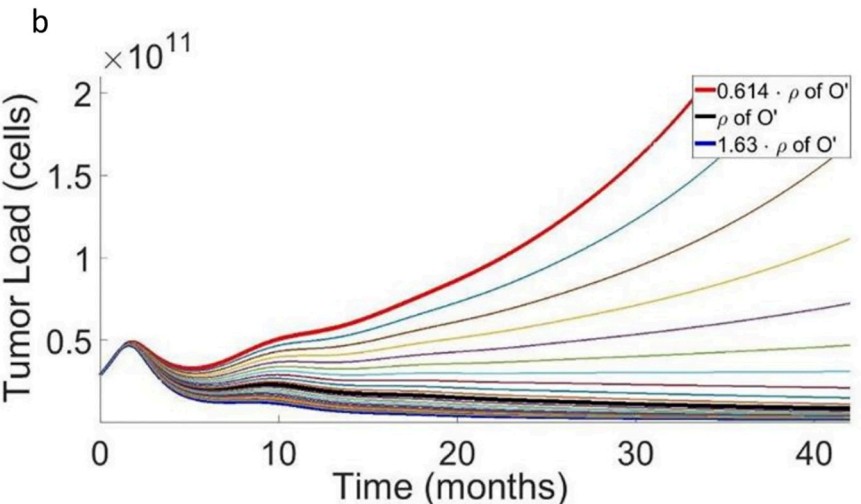

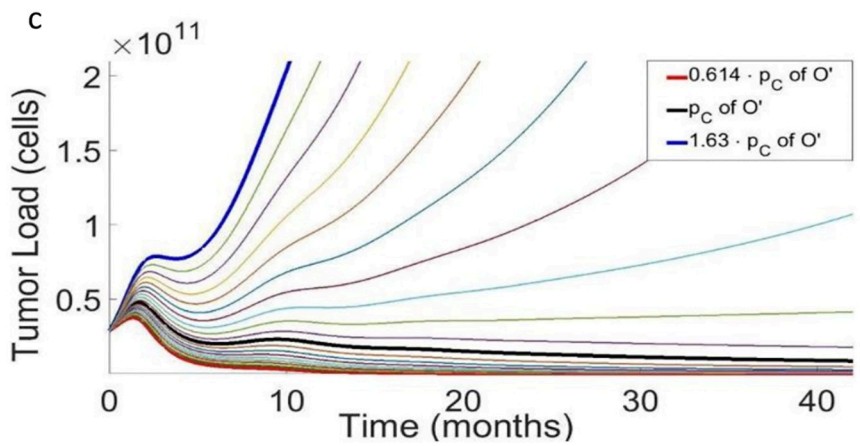

**Fig 3. Tumor progression as a function of the patients' personal parameters.** (3a) Simulation results of 20 hypothetical patients treated by pembrolizumab, 120 mg, and every 3 weeks. The simulated patients are similar in all parameters, except for the cytotoxicity of effector $CD8^+$ T cells to cancer cells, $k_E^0$, whose values for 3 different patients are indicated in the color legends, embedded in the figure. For all other parameters, see caption to Fig 2 and the Materials and Methods section. (3b) Simulation results of 20 hypothetical patients treated by pembrolizumab, 120 mg, every 3 weeks. The simulated patients have the same parameters, except for tumor immunogenicity, $\rho_D$, whose values for 3 different patients are indicated in the color legends embedded in the figure. For all other parameters, see caption to Fig 1 and the Materials and Methods section. (3c) Simulation results of 20 hypothetical patients treated by pembrolizumab, 120 mg, every 3 weeks. All the simulated patients have similar parameter values, except for the tumor growth rate, $p_C$, whose values for 3 different hypothetical patients are indicated in the color legends embedded in the figure. For all other parameters, see caption to Fig 1 and the Materials and Methods section.

The parameters $k_C^{03}$ and $k_C^{04}$ are the baseline measures of the effects of the cancer tumor on the cytotoxicity of EM and E T cells, respectively. In other words, $k_C^{03}$ and $k_C^{04}$, stand for the effect of PD-L1 ligand found on tumor cells when bound to PD1 receptors on EM and E cells. We simulated the response to pembrolizumab, 120 mg, every 3 weeks for 28 months, for values of $k_C^{03}$ and $k_C^{04}$, varying within a 1.63 fold range above and below their respective estimated baseline value for Patient O'. The simulated results show that the influence of $k_C^{04}$ is comparable to that of $k_E^0$, while that of $k_C^{03}$ is of a lesser effect (in the neighborhood of the best-fit point, see below). Thus, large $k_C^{04}$ values yield a monotonic increase in tumor load; most intermediate values show a non-monotonic change in tumor load, having a local maximum in the first year of treatment and different rates of disease thereafter. Finally, our model simulations suggest that patients with low $k_C^{03}$, $k_C^{04}$ values should manifest a CR, as expected intuitively (not shown).

The parameter $p_c$ represents the intrinsic cancer cell division rate. Its effect on tumor progression in a melanoma patient is shown in Fig 3C, where large values yield progressive disease and small values yield CR after about two years of treatment. Note that "disease flare" or HPD, namely, a rapidly progressing disease, was not retrieved by these simulations: a temporary decline in the rate of disease progression within the first few months following immunotherapy onset can be seen even at high cancer cell division rates.

## Can the ratio between the reinvigoration rate of CD8+ T cells and tumor load at baseline predict the response to pembrolizumab?

Huang et al., suggest that the ratio between the reinvigoration of circulating CD8+T cells and the baseline tumor load positively correlates with clinical response to pembrolizumab and can serve as a response predictor for this drug [33]. We tested our model's feasibility by using it to retrieve this result. To achieve this goal, we generated four virtual patient populations (denoted VP0-VP3) in which all model parameters were taken as constant except for one or several of those shown above to potentially affect the patient-specific response. The latter varied in plausible ranges and the response to treatment of each virtual patient was monitored according to irRECIST criteria [50]. The results, summarized in Fig 4 and Table 1, show a good classification accuracy of responders/non-responders when both effector toxicity and cancer cell replication rate are determined personally, and when baseline tumor load is associated with replication rate of cancer cells. The method for performing the classification and determining its accuracy is detailed in the Supporting Information file.

## Discussion

In this work, we showed theoretically that a rapid increase in tumor load in patients with advanced melanoma, soon after the onset of immunotherapy by pembrolizumab, can be due to relatively low toxicity of effector CD8+ T cells to cancer cells. More generally, we showed

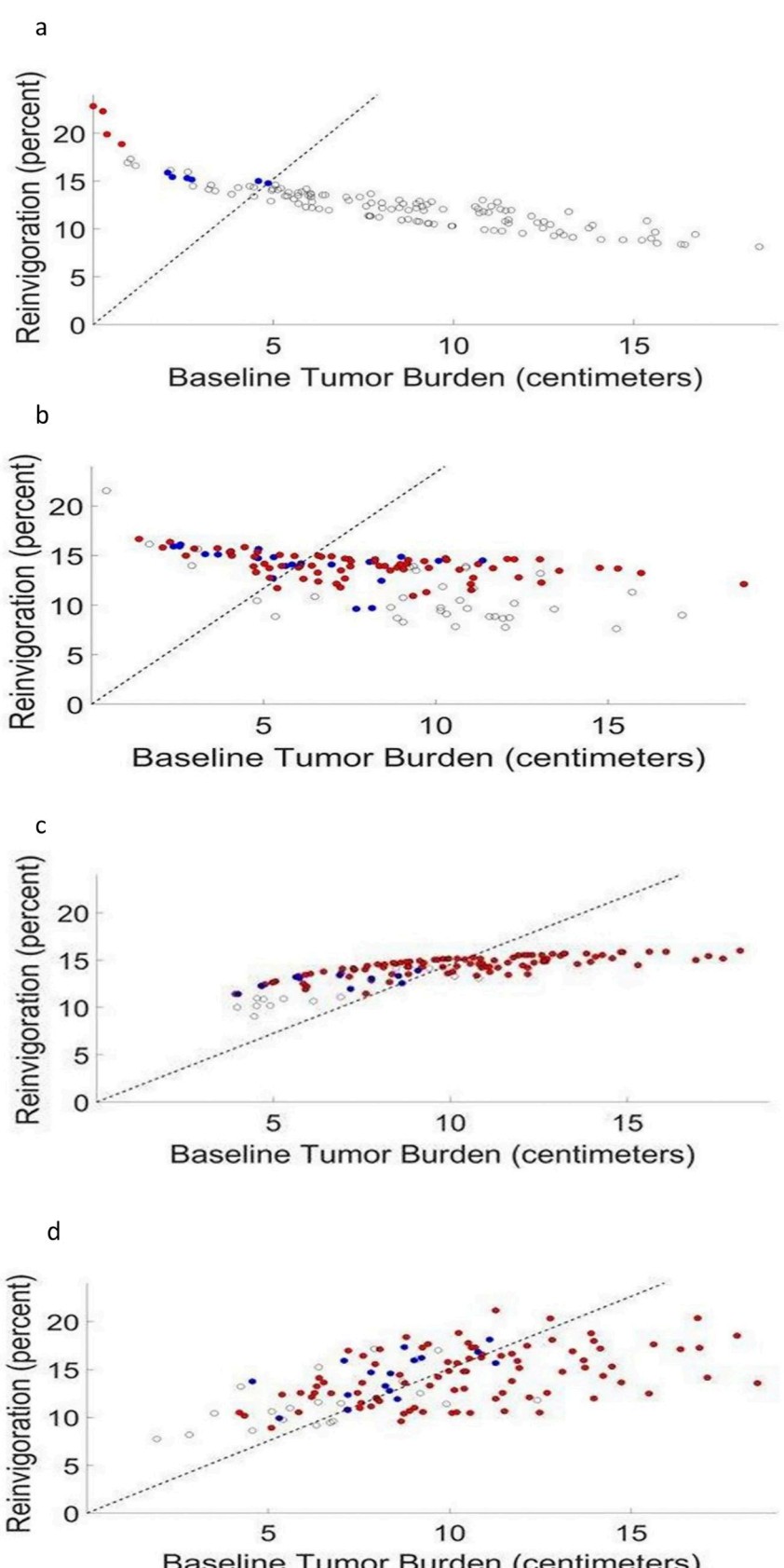

**Fig 4. Response to ICI by pembrolizumab as a function of reinvigoration rate and tumor load at the onset of immunotherapy.** Four different patient populations, under treatment by pembrolizumab, were simulated for retrieving the clinical trial in [33]. Each population included a hundred and twenty-five virtual patients who varied in baseline tumor load and number of lesions; some of the populations also varied in one or two other parameters. Each circle represents a different patient, mapped on the abscissa according to his/her tumor load measured at immunotherapy onset, and on the ordinate according to its measure of reinvigoration, i.e. the percentage of Ki67 +CD8$^+$ T cells, as evaluated by the number of cells whose division rate became and remained positive during the first six weeks of treatment (see Materials and Methods section for details). The colors stand for the patient's response: complete response (CR; not obtained in our simulations), partial response (PR; blue), stable disease (SD; empty black circles) and progressive disease (red). (4a) patients differ in their initial tumor load and number of lesions, and are identical in all other disease and immune response parameters. (4b) the toxicity of effector T cells toward tumor cells, $k_E^0$, and the cell division rate of cancer cells, $p_C$, are patient-specific, i.e., randomly selected from the pre-defined ranges, 0.28 to 3.5 of a heuristically determined base value (see Materials and Methods). (4c) as in 4a, except that the cell division rate of cancer cells, $p_C$, is now assumed to be strictly correlated with the baseline tumor load. (4d) as in 4c, except that the cell division rate of cancer cells, $p_i$, are also randomly selected from the pre-defined ranges of 0.5 to 2 of their respective base values.

how a relatively small patient variation in immune or disease parameters could lead to a large variation in the response to ICI. We also confirmed the potential utility of the ratio [reinvigoration/tumor load] as a response predictor [33]. Our study was initiated by developing a new mechanistic model for the interactive dynamics of the cellular arm of the immune system, the advanced melanoma cancer and the ICI drug, pembrolizumab. This model is new in taking into account the specific roles of T memory stem cells within the dynamic differentiation of

**Table 1. Properties and response of the virtual populations.**

| Virtual Population Name | Patient-specific Parameters[1] | Results | FIGURE |
|---|---|---|---|
| VP0 | Baseline tumor load, number of metastases. | 1. Lower reinvigoration is associated with larger initial tumor load.* | 4a |
| VP1 | As in VP0, plus toxicity of effector T cells toward tumor cells, $k_E^0$, and intrinsic division rate of cancer cells, $p_c$. | 1. Weighted classification accuracy of 0.69—clusters of responders/non-responders are considerably mixed;** 2. Low reinvigoration (corresponding to low $k_E^0$ and $p_c$ values) result in SD only, regardless of initial tumor load. 3. A negative correlation between reinvigoration and initial tumor load, but less negative than that of VP0. | 4b |
| VP2 | As in VP1, however, in this population $p_c$ is completely correlated with the baseline tumor load. | 1. High classification accuracy of 0.75—a good distinction between responders and non-responders clusters; 2. Lower reinvigoration results in SD only; 3. A positive correlation between the CD8$^+$ T cells reinvigoration and baseline tumor load;*** 4. Range of reinvigoration levels is significantly narrower than in the clinical cohort; 5. A significant correlation between partial response (PR) and lower baseline tumors.**** | 4c |
| VP3 | As in VP2, plus intrinsic division rates of the developing CD8$^+$ T cells, $p_i$. | 1. Classification accuracy of 0.72—a good distinction between responders and non-responders clusters; 2. A higher positive correlation between initial tumor load and reinvigoration level than that in VP2; 3. The reinvigoration range is comparable to the clinical findings [33]. | 4d |

[1]All other parameters are the same in all virtual populations

* A trend opposite to the one observed in Huang et al., [33].

** Compare to the weighted classification accuracy of 0.77 in Huang et al.'s cohort (based on Fig 9B in [33]). For calculation of classification accuracy, see the Materials and Methods section

*** In agreement with Fig 9B in [33].

**** In some contradiction to [33], where the response is found to be correlated to the reinvigoration to baseline tumor ratio, but not to any of them separately.

lymphocytes, as suggested by Gattinoni et al. [23]. The model is also new in accounting for both senescence and exhaustion in the cellular immune arm—two dominant mechanisms in the context of PD-L1/PD-1 checkpoint blockade immunotherapy [12, 51].

Model simulations successfully captured the atypical disease dynamics in an individual patient, as exemplified by the retrieval of the complex clinical response pattern of Patient O' (**Fig 2** and **Table A in S1 Text**). Both data and simulated dynamics display a significant increase in tumor load, with a maximum, seemingly at the fifth week of therapy, pursued by shrinkage of tumor load much below the load at baseline, another increase to a maximum at the forty fifth week of therapy, and a slow, long-lasting, descent in tumor load to less than 30% of baseline load after 162 weeks of therapy. The response pattern at around the forty-fifth week of treatment bears the characteristics of a delayed pseudoprogression, clinically defined as a ≥ 25% increase in tumor load at any assessment after week 12, which is not confirmed as a progressive disease at the subsequent imaging assessment [5, 50]. The clinical record of Patient O' at that time appeared as a mixed response, including small shrinkage of the target lesion, but the growth of lesions in other organs (**Table A in S1 Text**). Because it looked like progression on scans, the decision to continue the pembrolizumab treatment was not straightforward at the time. *A posteriori*, it appeared like the right decision since the tumors of this patient further shrank and have continued to decrease more than forty months after pembrolizumab onset (personal information; **Fig 2**, and **Table A** in **S1 Text**). Following validation of the model presented here, one will be able to use it to aid clinical decision-making for specific patients, as was theoretically suggested for immunotherapy [20], or performed in combined chemo/biological therapy [52].

The specific model parameters of Patient O' were evaluated to serve as a reference for further analysis, and their reliability was verified. Subsequently, we simulated the model over diverse parameter values around those of Patient O's model. Results show that inter-patient variation in the values of only one parameter, most notably, T cell toxicity, suffices for yielding a rich variation in response, including a pattern which can roughly fit the description of HPD —an acceleration in tumor growth rate—whose numerical definition varies among authors [10, 11], [53–55]. Champiat et al., [11], and Chubachi et al., [12], were first to point to HPD or "disease flare" as a response associated with anti-PD1/PD-L1 treatment and to raise a concern about the use of ICI. Analyzing a small group of patients, Kato et al., [10] infer that some patients suffer from ICI-driven HPD, and suggest several potential molecular mechanisms to account for this phenomenon. In spite of the low frequency of HPD among the ICI-treated melanoma patients (~10%; [10, 11]), and the insufficient knowledge about the underlying mechanism, the possibility that treatment by ICI can aggravate the disease is a serious worry to many oncologists.

Based on our results, we anticipate that low cytotoxicity of effector CD8+ T cells significantly reduces the efficacy of ICI drugs. This effect, e.g., due to advanced age [56] is expected to result in accelerated tumor progression, i.e., HPD. We hypothesize that the cytotoxicity parameter of effector cells can be a potential response marker in advanced melanoma patients. This possibility should be tested in the patient population, possibly, by using the molecule CD56—a phenotypic marker bearing strong immuno-stimulatory effector functions [57]. We believe that in patients who are prone to 'disease flare' due to limited T cell cytotoxicity, disease deterioration can be limited by the replacement of ICI by nonimmune-related therapeutic modalities.

In this work, we employed the concept of the virtual clinical trial (see above) to retrieve and analyze the clinical trial of Huang et al., [33], showing that in melanoma, the ratio of reinvigoration rate to baseline tumor load can be used to cluster patients according to the quality of their response. Our methodology involved the creation of plural virtual patient populations

that differed in specific assumptions about the parametric relationships, and the use of these virtual populations to perform virtual trials which resemble those of Huang et al. This methodology was instrumental in deciphering the crucial intrinsic properties which underlie the character of the reinvigoration/basic tumor load ratio in [33]. Taken together, our virtual clinical trials, simulating the "true" clinical trial in [33], further confirm the validity of our model and demonstrate how different decisive personal parameters can be combined in a complex manner into one quantifiable measure, which can serve to classify responders.

The mechanism of pseudoprogression has yet to be fully understood [58]. Our model simulations imply that a temporary increase in total tumor load is unlikely to result from real tumor growth and shrinkage thereafter, as may be implied, e.g., [13]: exhaustive computational analysis of our model could not retrieve Patient O' disease dynamics under the assumptions of progressively decreasing self-renewal probabilities along the T cell developmental pipeline, as suggested by Gattinoni et al., [23–25]. Instead, in our simulations, true short-term tumor growth was obtained only when assuming aberrant self-renewal probabilities of T cells, acting as a buffer or as a delay mechanism. This assumption was necessary for retrieving the complex dynamics, regardless of other parameter values in the model. We infer, therefore, that in real life, pseudoprogression cannot result from true changes in the total tumor volume. Alternatively, since the disease of Patient O' is atypical [28], it is not implausible that her equally uncommon disease dynamics reflect some aberrations in her T cell self-renewal capacity. Further research on this aspect is warranted.

In conclusion, our model was useful in analyzing clinical phenomena related to advanced melanoma treatment by ICI. The simulation results show that one cannot rely on simple differences between patients in the expression of PD-L1 on the surface of tumor cells, on PD-1 on the surface of T cells, or the tumor immunogenicity, for predicting response to pembrolizumab. Other characteristics of the patient's immunity, notably, cytotoxicity of effector T cells, have to be evaluated to enable prognosis. This can be aided by the use of a mathematical mechanistic model that processes the pertinent complex information and rationalizes the relationships between the molecular and the clinical parameters. Following validation by a new data set of pembrolizumab-treated advanced melanoma patients, our model will hopefully serve for immunotherapy personalization.

## Supporting information

**S1 Text. Detailed mathematical modeling, parametrization, generation, and analysis of virtual populations.** Table A. Clinical Information for Patient O'
Table B. Values and References for the Generic Model Parameters
Fig A. Sensitivity analysis of Patient O's model for the parameter $p_c$
Fig B. Sensitivity analysis of Patient O's model for the parameter $k_C^{03}$
(DOCX)

## Acknowledgments

We thank Marina Kleiman and Abir Abu Koider for the collection of the clinical data, and Dorit Dror for administrative assistance.

## Author Contributions

**Conceptualization:** O. Shlagman, Z. Agur.

**Data curation:** A. Yakobson, I. Lazarev.

**Formal analysis:** D. Perlstein.

**Investigation:** K. Halevi-Tobias.

**Methodology:** D. Perlstein, O. Shlagman, Y. Kogan, K. Halevi-Tobias, Z. Agur.

**Resources:** A. Yakobson, I. Lazarev.

**Software:** D. Perlstein.

**Supervision:** Y. Kogan, Z. Agur.

**Writing – original draft:** Z. Agur.

**Writing – review & editing:** D. Perlstein, Y. Kogan, I. Lazarev, Z. Agur.

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
