## [Decision Letter · Decision Letter 0]

15 Oct 2019

PONE-D-19-25008

Personal response to immune checkpoint inhibitors of patients with advanced melanoma explained by a computational model of cellular immunity, tumor growth, and drug

PLOS ONE

Dear Dr. Agur,

Thank you for submitting your manuscript to PLOS ONE. After careful consideration, we feel that it has merit but does not fully meet PLOS ONE’s publication criteria as it currently stands. Therefore, we invite you to submit a revised version of the manuscript that addresses the points raised during the review process.

We would appreciate receiving your revised manuscript by Nov 29 2019 11:59PM. To enhance the reproducibility of your results, we recommend that if applicable you deposit your laboratory protocols in protocols.io, where a protocol can be assigned its own identifier (DOI) such that it can be cited independently in the future. For instructions see: http://journals.plos.org/plosone/s/submission-guidelines#loc-laboratory-protocols

We look forward to receiving your revised manuscript.

Kind regards,

Filippo Castiglione

Academic Editor

PLOS ONE

Journal Requirements:

Additional Editor Comments (if provided):

Reviewers' comments:

Reviewer's Responses to Questions

**Comments to the Author**

1. Is the manuscript technically sound, and do the data support the conclusions?

Reviewer #1: Yes

Reviewer #2: Yes

2. Has the statistical analysis been performed appropriately and rigorously? 

Reviewer #1: No

Reviewer #2: I Don't Know

3. Have the authors made all data underlying the findings in their manuscript fully available?

Reviewer #1: Yes

Reviewer #2: No

4. Is the manuscript presented in an intelligible fashion and written in standard English?

Reviewer #1: Yes

Reviewer #2: No

5. Review Comments to the Author

Reviewer #1: Minor comments:

- Regarding the parameters' estimation, authors used a simulation based method, that is the

estimated parameters are those that minimise the distance between the observed tumor load

and the one obtained from the simulations. The minimisation has been carried-out through

an \\algorithm that combines various scheme and search methods...". It would be worthwhile

if authors could give more details regarding how do they implemented such algorithm together

with a quantitative measure regarding the eciency gained compared to the built-in matlab

algorithms.

-Regarding the sensitivity analysis, the authors varied the parameters one by one in a given

range and looked at the behaviour of the cost function. This approach does not take into

account the multidimensional nature of the cost function, therefore the authors should evaluate

the cost function over a multidimensional grid in which all the parameters vary simultaneously.

-Generally, point estimation provides poor information because it could vary with the

simulations. For this reason, authors should provide estimated condence intervals instead

of just the single point estimation. Condence intervals would give information regarding the

robustness of the parameters' estimate together with an hint regarding the range in which the

parameters should vary.

-It is not clear why the authors choose to vary the parameters in 1:63-fold range above and

below the estimated baseline values, maybe this choice should be discussed a bit.

-Regarding the classication between responders and non-responders obtained in the 4 virtual

populations, the plots in gure 4 do not show an evident clustering of the two groups but

Table 1 gives a description of the obtained results. However, the method adopted to evaluate

the classication accuracy is not given, therefore it would be worthwhile to add more details.

Please find attached all the comments and typos.

Reviewer #2: Summary:

The authors have developed a mathematical/computational model to capture the dynamics of within-host progression of melanoma in the context of a particular monoclonal antibody treatment, pembrolizumab. The main question they are addressing is whether their model reflects important features of disease progression and response to treatment in a way that can be personalized to an individual. Checkpoint inhibitors are a relatively new treatment modality, so it is timely to be modeling such a treatment.

The mathematical model is a system of ordinary differential equations, so cell concentrations are being tracked, and spatial features of melanoma are not considered. This is a reasonable modeling decision. The model is trained and parameters are determined via data collected from a single patient. Using the model thus trained, the authors then simulate a virtual population of patients by varying the model parameters. The virtual patient population is able to capture overall response rates given in a particular clinical study.

The authors conclude that, based on model simulations, tumor progression can result from a weakened patient immune response, and they provide a hypothesis as to how predict an individual's response to treatment with pembrolizumab. The authors state that further model validation is needed, but that this is a first step toward the possibility of personalizing immunotherapy protocols. If this approach could be used to personalize future treatment protocols, it would be a very important contribution.

Comments (General):

The authors state, "Simulations of virtual populations were in good agreement with clinical results." It was not immediately apparent which clinical results are being used for comparison data. The first reading of this seems to imply that there will be a comparison to clinical trial population level data, but it is difficult to see where that is done.

For the sake of validation, is there a clinical study against which the virtual population outcomes can be compared?

The approach of creating a within-host model and then simulating a population using that model is a good one, and has been implemented in other works. It would help the reader if the authors were to cite other works that have taken a similar approach, and clarify in what ways their approach is similar or differs. Individual to population simulation work has been done in, for example, http://www.sciencedomain.org/abstract/3987.

The model may have potential for personalizing this MAB treatment approach, and further validation would be crucial. To that end, if readers can experiment with the model, testing and validation could happen more rapidly. There should be sufficient information provided to allow the model to be fairly easily implemented and simulation results reproduced by an interested reader. The authors state that the simulations were run in Matlab. It would be very helpful if the authors made the code both for the simulations and for the parameter fitting schemes available for testing purposes.

Comments (Minor):

In Figure 1a, the authors state that "The cancer cells (khaki ellipses) divide by a power-law

function, whose exponent is 2/3." A citation to the power-law growth model is provided later in line 205. The citation is a study done in breast cancer, yet the cancer being modeled in this paper is melanoma. Since it has been observed that different cancer types may grow according to different intrinsic growth laws, wt would help the reader if there were some further explanation as to why the power law, with exponent 2/3, was selected (as opposed to any other "popular" intrinsic growth law, such as Gompertzian or logistic).

Further clarification of the biological background would go a long way toward help a non-specialist audience appreciate this work. For example, it would help to further clarify the background provided on this treatment approach. Background is already provided by the authors, but the discussion could be clearer. An example of an improved explanation of what PD-1 checkpoints are and why checkpoint inhibitors are a treatment approach of interest can be found here:

https://www.ncbi.nlm.nih.gov/pmc/articles/PMC6129935/

That abstract states:

"In the tumor microenvironment, overexpression of check point molecules, such as program death ligand-1 (PD-L1) functions as a protector against the immune surveillance by T-cell through the interaction with program death-1 (PD-1) receptor (1). It has been found that PD-1 overexpressed on a variety of tumor-killing immune cells, such as monocytes, macrophages, cytotoxic CD8+ T cells, and dendritic cells has an active role in hijacking the antitumor immune response (2). Therefore, inhibition of PD-1 and PD-L1 interaction would resurrect the tumor-killing effect of CD8+ T cells (3,4). The immunosuppressive regulatory T cells (Treg, CD4+ Foxp3+) also overexpress PD-1 receptor that favors immune suppression of tumor and negatively regulates CD8+ T cells (5,6). Thus, the use of PD-1 inhibitors not only reactivate the function of CD8+ T cell but also downmodulate the function of Treg and tumor-associated macrophage (TAM) cells through inhibition of mammalian target of rapamycin (mTOR)-Akt and Stat3 signaling cascade (7) (Figure 1). Several clinical trials are ongoing using PD-1/PD-L1 therapies for NSCLC (e.g., nivolumab, pembrolizumab, atezolizumab, and durvalumab) (Figure 2). The use of immunotherapy has been a game changer in comparison to conventional cancer therapy as they can be personalized for individual therapy or can be combined with chemotherapeutics, targeted therapeutics and nanomedicines (9-13)."

A straightforward definition of pembrolizumab up front would also help. For example, from https://www.cancer.gov/publications/dictionaries/cancer-terms/def/pembrolizumab:

Pembrolizumab binds to a protein called PD-1, which is found on T cells. Pembrolizumab may block PD-1 and help the immune system kill cancer cells. It is a type of monoclonal antibody and a type of immune checkpoint inhibitor. Also called Keytruda.

Figure 1a. Some elements of the figure are so closely overlapping that it makes the figure difficult to read.

6. PLOS authors have the option to publish the peer review history of their article (what does this mean?). If published, this will include your full peer review and any attached files.

Reviewer #1: No

Reviewer #2: No

---

## [Author Response · Author response to Decision Letter 0]

21 Nov 2019

Please see the attached files, including Reply to Reviewers.

---

## [Decision Letter · Decision Letter 1]

10 Dec 2019

Personal response to immune checkpoint inhibitors of patients with advanced melanoma explained by a computational model of cellular immunity, tumor growth, and drug

PONE-D-19-25008R1

Dear Dr. Agur,

We are pleased to inform you that your manuscript has been judged scientifically suitable for publication and will be formally accepted for publication once it complies with all outstanding technical requirements.

With kind regards,

Filippo Castiglione

Academic Editor

PLOS ONE

Additional Editor Comments (optional):

Reviewers' comments:

Reviewer's Responses to Questions

**Comments to the Author**

1. If the authors have adequately addressed your comments raised in a previous round of review and you feel that this manuscript is now acceptable for publication, you may indicate that here to bypass the “Comments to the Author” section, enter your conflict of interest statement in the “Confidential to Editor” section, and submit your "Accept" recommendation.

Reviewer #1: All comments have been addressed

Reviewer #2: All comments have been addressed

2. Is the manuscript technically sound, and do the data support the conclusions?

Reviewer #1: Yes

Reviewer #2: Yes

3. Has the statistical analysis been performed appropriately and rigorously? 

Reviewer #1: Yes

Reviewer #2: Yes

4. Have the authors made all data underlying the findings in their manuscript fully available?

Reviewer #1: Yes

Reviewer #2: Yes

5. Is the manuscript presented in an intelligible fashion and written in standard English?

Reviewer #1: Yes

Reviewer #2: Yes

6. Review Comments to the Author

Reviewer #1: The authors fully provide details and integrations regarding issues raised in the previous round of review.

Reviewer #2: The authors have addressed the areas of concern. The added information helps the reading flow more smoothly.

7. PLOS authors have the option to publish the peer review history of their article (what does this mean?). If published, this will include your full peer review and any attached files.

Reviewer #1: No

Reviewer #2: No

---

## [Editor Report · Acceptance letter]

16 Dec 2019

PONE-D-19-25008R1 

Personal response to immune checkpoint inhibitors of patients with advanced melanoma explained by a computational model of cellular immunity, tumor growth, and drug 

Dear Dr. Agur:

I am pleased to inform you that your manuscript has been deemed suitable for publication in PLOS ONE. Congratulations! Your manuscript is now with our production department. 

With kind regards,

on behalf of

Dr. Filippo Castiglione 

Academic Editor

PLOS ONE